# A cluster randomised controlled trial to evaluate the effectiveness and cost-effectiveness of the GoActive intervention to increase physical activity among adolescents aged 13–14 years

Helen Elizabeth Brown,[1] Fiona Whittle,[1] Stephanie T Jong,[1] Caroline Croxson,[2] Stephen J Sharp,[1] Paul Wilkinson,[3] Edward CF Wilson,[4] Esther MF van Sluijs,[1] Anna Vignoles,[5] Kirsten Corder[1]

► Prepublication history and additional material are available. To view these files, please visit the journal online (http://dx.doi.org/10.1136/bmjopen-2016-014419).

[1]UKCRC Centre for Diet and Activity Research (CEDAR) and MRC Epidemiology Unit, University of Cambridge, Cambridge, UK
[2]Nuffield Department of Primary Care Health Sciences, University of Oxford, Oxford, UK
[3]Department of Psychiatry, University of Cambridge, Cambridge, UK
[4]Cambridge Centre for Health Services Research, University of Cambridge, Cambridge, UK
[5]Faculty of Education, University of Cambridge, Cambridge, UK

**Correspondence to**
Dr Helen Elizabeth Brown;
heb56@medschl.cam.ac.uk

## ABSTRACT

**Introduction** Adolescent physical activity promotion is rarely effective, despite adolescence being critical for preventing physical activity decline. Low adolescent physical activity is likely to last into adulthood, increasing health risks. The Get Others Active (GoActive) intervention is evidence-based and was developed iteratively with adolescents and teachers. This intervention aims to increase physical activity through increased peer support, self-efficacy, group cohesion, self-esteem and friendship quality, and is implemented using a tiered-leadership system. We previously established feasibility in one school and conducted a pilot randomised controlled trial (RCT) in three schools.

**Methods and analysis** We will conduct a school-based cluster RCT (CRCT) in 16 secondary schools targeting all year 9 students (n=2400). In eight schools, GoActive will run for two terms: weekly facilitation support from a council-funded intervention facilitator will be offered in term 1, with more distant support in term 2. Tutor groups choose two weekly activities, encouraged by older adolescent mentors and weekly peer leaders. Students gain points for trying new activities; points are entered into a between-class competition. Outcomes will be assessed at baseline, interim (week 6), postintervention (week 14–16) and 10-month follow-up (main outcome). The primary outcome will be change from baseline in daily accelerometer-assessed moderate-to-vigorous physical activity. Secondary outcomes include accelerometer-assessed activity intensities on weekdays/weekends; self-reported physical activity and psychosocial outcomes; cost-effectiveness and cost-utility analyses; mixed-methods process evaluation integrating information from focus groups and participation logs/questionnaires.

**Ethics and dissemination** Ethical approval for the conduct of the study was gained from the University of Cambridge Psychology Research Ethics Committee. Given the lack of rigorously evaluated interventions, and the inclusion of objective measurement of physical activity, long-term follow-up and testing of causal pathways, the results of a CRCT of the effectiveness and cost-effectiveness of GoActive are expected to add substantially

## Strengths and limitations of this study

► The strengths of the GoActive evaluation study include the cluster randomised controlled trial design, objective measurement of physical activity, long-term follow-up and testing of causal pathways to rigorously assess the effectiveness and cost-effectiveness of the GoActive programme.

► We will recruit 16 secondary schools from both Essex and Cambridgeshire. A possible limitation of the study is that, despite our purposive sampling of schools with varied socioeconomic status, it is likely that participants may not be entirely representative of the wider UK population (particularly with regards to ethnicity).

to the limited evidence on adolescent physical activity promotion. Workshops will be held with key stakeholders including students, parents, teachers, school governors and government representatives to discuss plans for wider dissemination of the intervention.

**Trial registration number** ISRCTN31583496.

## BACKGROUND

Physical activity is protective against obesity and related metabolic disorders in young people.[1 2] Meta-analytic data from 20 871 4–18-year-olds suggest that every 10 min increase in moderate-to-vigorous activity (MVPA) is associated with a smaller waist circumference (−0.52 cm) and lower fasting insulin (−0.028 pmol/L).[2] In adolescence, physical activity declines 7% per year.[3] Low physical activity in adolescence is also likely to progress to adulthood inactivity,[4] increasing the risk of diabetes, cancer and mortality.[5 6] Adolescence is therefore a critical period to increase physical activity,[7] both due to the aforementioned decline and because pubertal, brain and social development during this time leads to new capacity for changing

health behaviours,[8] increasing the likelihood of long-term change.

The 2012 Chief Medical Officer's report states the importance of physical activity among young people[9] and a recent international expert panel concluded that developing effective and sustainable interventions to increase physical activity among young people is the most important priority in the physical activity research field.[10] Further, the recently published report from the All-Party Commission on Physical Activity calls specifically for the creation of active schools, including the provision of a more diverse and inclusive offer of physical activity.[11]

Reviews highlight the limited efficacy of existing adolescent physical activity promotion interventions.[12–15] We have previously identified several possible reasons for this lack of effectiveness[16]; for example, many interventions only target subgroups (such as girls[17] or low socioeconomic groups)[18] despite activity declining among all groups.[16] We aim to recruit the whole school year group for evaluation, and to target all groups in the Get Others Active (GoActive) intervention, which to our knowledge has rarely been done in physical activity promotion interventions. In addition, the decline in activity mainly occurs out of school[16]; however, many interventions only target specific school-based times; for example, school time[13 19] or Physical Education (PE) lessons,[20] whereas GoActive encourages participants to do more activity both in and out of school. Further, very few adolescent physical activity interventions, especially among older adolescents, have been evaluated using objective measurement of physical activity,[14] and include long-term follow-up, process evaluation or an assessment of cost-effectiveness.[21] This therefore highlights an urgent need for more rigorous evaluation of potentially effective strategies to increase physical activity in adolescents.

### Objectives
The primary aim of this study is to assess the 10-month effectiveness of the GoActive intervention to increase average daily objectively measured MVPA among 13–14-year-old adolescents. We will also assess the effect of GoActive immediately postintervention, and on the following secondary outcomes: (1) objectively assessed activity intensities during school time, weekday evenings and weekends; (2) student-reported physical activity participation, self-efficacy, peer support, social networks, self-esteem, friendship quality (proposed mediators) and well-being, and school-level attendance and academic performance; and (3) body composition (body fat percentage and body mass index (BMI) z-score). We will investigate potential moderation of intervention effects by sex, socioeconomic status, ethnicity, baseline activity level and weight status, and potential mechanisms of effect by proposed mediators using a mixed-methods approach. Further, we will assess short-term (within-trial) and potential long-term cost-effectiveness of the GoActive intervention and will conduct a comprehensive process evaluation including questionnaires, focus groups, and individual interviews (with participants,

mentors, teachers, and intervention facilitators), data from intervention logs and website analytics.

## INTERVENTION
The development of the GoActive intervention with supporting rationale has been described in detail previously.[22] Briefly, each year 9 class (tutor group or home room class) chooses two activities each week from a selection provided. There are currently 20 activities available, using little or no equipment, and appealing to a wide variety of students (including Ultimate Frisbee, Zumba and Hula Hoop). Materials available on the password-protected GoActive intervention website include activity instructions (Quick Cards) which offer an overview of each activity, a short explanation, suggestions for adaptations and provide advice, safety tips and 'factoids', in addition to a short video introducing each activity. GoActive is implemented using a tiered-leadership system where mentors (older adolescents within the school) and peer leaders (within each year 9 class) encourage students to try these activities each week. The mentors remain paired with each class for the duration of the intervention, whereas the peer leaders (two per class each week, one male and one female) change every week. In addition to the student leaders, a local authority-funded intervention facilitator will support the programme during the first term of delivery and will provide distant support thereafter.

Teachers are encouraged to use one tutor time weekly to do one of the chosen activities as a class; however, students gain points for trying these new activities at any time in or out of school. Points are gained every time they try an activity; there is no expectation of time spent doing the activity as points are rewarded for the taking part itself. Individual students keep track of their own points privately on the study website and their points are entered into the between-class competition. Class rankings are available on the website to encourage teacher support and students receive small rewards (such as a sports bag, t-shirt, or hoodie) for reaching individual points thresholds.

## METHODS
### Study design
We will conduct a school-based cluster randomised controlled trial (CRCT) of the GoActive intervention. The study will be conducted in government-funded, non-fee-paying (state), all-ability, co-educational secondary schools including year 9 students in Cambridgeshire and Essex, UK. After baseline measurements (September–December 2016), schools will be randomly allocated to one of two conditions: (1) to deliver the GoActive intervention to the whole of year 9 or (2) to a no-treatment control group. Participant data collection will occur at baseline, 6 weeks, 14–16 weeks and 10 months (primary outcome). The protocol will be conducted and reported in accordance

with Standard Protocol Items: Recommendations for Interventional Trials (SPIRIT) guidance (see online supplementary data 2).[23–25]

## RECRUITMENT PROCEDURES

### Schools

We will recruit 16 secondary schools with a mixture of socioeconomic status, representative of UK variability. Head teachers, year 9 leaders and PE leaders from all eligible schools will be sent an invitation letter and school information sheet via email. These documents will describe the study procedures (eg, student recruitment and consent, measurements) and will include an electronic link to an information video describing GoActive. A follow-up phone call to each school will be made approximately 1 week after the initial invitation, asking for a meeting with relevant staff to discuss the study and request consent to participate. Phone calls and repeat emails will continue until 16 schools (8 in Cambridgeshire and 8 in Essex) have provided consent to participate. We will also create a waiting list to replace any schools who may withdraw from the study prior to randomisation. We will also use our existing networks and school contacts to facilitate school recruitment. Schools that do not agree to take part will be asked to select the most relevant reason for their refusal from a predetermined list (eg, lack of interest, lack of time).

### Participants

All year 9 students (13–14 years) in participating schools will be eligible to participate in study measurements. As in feasibility and pilot work, we plan to include participants with a disability and those with learning or movement difficulties, taking care to follow advice from schools.[26] This is appropriate due to the inclusive nature of the GoActive intervention and will help to avoid stigmatisation of any groups within schools.[27] As such, no exclusion criteria will be applied.

All year 9 students and their parents will receive a paper invitation pack, including a participant information sheet and an invitation to participate in study measurements. These information packs will be distributed to students during an introductory assembly conducted by a member of the GoActive team; students will be asked to take the packs home to their parents. Parents will also be sent duplicate information via email ('ParentMail' or the appropriate equivalent system as agreed by the school). Parents will be asked to provide passive consent (active opt-out consent) for their child to take part in study measurements. We will give parents at least 2 weeks to respond (a final date for response will be included in all correspondence). After 1 week, parents will receive an additional copy to ensure further opportunity for opting out prior to study measurements. Parents will be given the option to phone or email the study team (in lieu of returning a form) to facilitate their ability to respond. Reminders will additionally be included in all relevant school media, including regular newsletters

sent from the school. Written assent will be obtained from the students by research assistants trained in Good Clinical Practice prior to any baseline measurements taking place. Approved consent forms will be available on the study website http://www.goactive-uk.com. Mentors and teachers will provide written consent or assent (for those older and younger than 16 years, respectively) to participate in process evaluation following the same procedures as study participants.

Parental opt-out responses ranged from 2 (<1%) to 18 (7%) in feasibility and pilot schools, with 72%–88% of eligible students assenting to participate.[26] Recruitment rates using this strategy are substantially higher than previous UK-based research in this age group using parental opt-in consent (23% of eligible participants).[7] Participants will be informed that they can discontinue all or any part of the study (either or both measurements and intervention) at any time at their or their parent/guardian's request.

## SCHOOL RANDOMISATION

Schools will be stratified based on Pupil Premium (proxy for socioeconomic status, below/above the county-specific median; for information: https://www.gov.uk/guidance/pupil-premium-information-for-schools-and-alternative-provision-settings) and county (ie, Cambridgeshire or Essex). Randomisation lists for each stratum will be prepared by a statistician, using Stata (ref: StataCorp. 2015. Stata Statistical Software: Release 14. College Station, Texas, USA: StataCorp), after baseline measurements are completed to ensure schools and participants are unaware of their group allocation at baseline. Eight schools will be randomised to deliver the GoActive intervention and eight to a no-treatment control condition. For measurements after randomisation, it will not be possible to blind participants to randomised allocation as the intervention schools will have received the GoActive intervention.

Measurement staff will be blinded to intervention condition throughout the study as they will be trained and work separately from those involved in intervention delivery. Process evaluation with measurement staff will examine the success of blinding.

### Control condition

The control group will receive no-treatment or 'usual care', and no intervention will be implemented. If we were to offer the control group the intervention after follow-up measures, it would prevent us from potentially assessing longer-term impact of the programme. As such, this study has no waitlist control condition.

### DATA COLLECTION

Measurements will be conducted at four time points by trained researchers (figure 1). The primary measure of intervention effectiveness will be change from baseline in accelerometer-measured average daily MVPA at 10-month

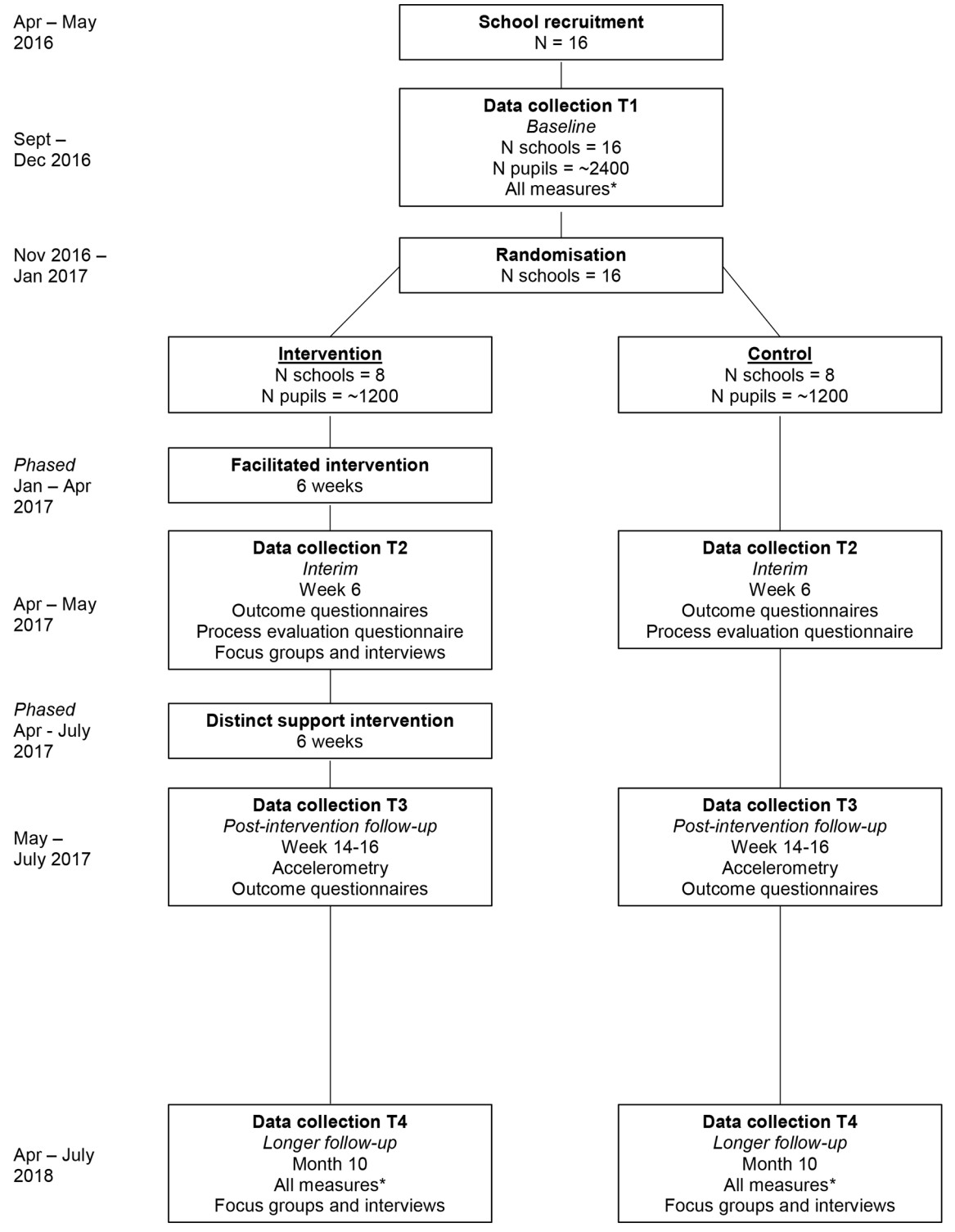

**Figure 1** Measurement sessions included in the GoActive evaluation.*All measures include accelerometry, anthropometry and outcomes questionnaire (student-reported physical activity participation, self-efficacy, peer support, group cohesion, self-esteem, friendship quality, and mood).

follow-up. All primary and secondary outcomes will be assessed at T1 and T4. Anthropometric measures will be removed from T3 (which will include all other outcomes, i.e. accelerometry and questionnaire-based measures), and T2 will focus on assessing the questionnaire-based measures only (including mediators of change). To prevent artificially inflated school-level clustering (due to weather

conditions or school events) and facilitate recruitment and retention, measurements at each school will be staggered over ≥2 weeks using a predetermined schedule.

### Accelerometry
The primary outcome will be accelerometer-assessed change in average daily MVPA between baseline and

10-month follow-up. Secondary accelerometry outcomes will be change from baseline in average minutes spent in sedentary and light activity, as well as overall physical activity (counts per minute) during school, weekdays after school and at weekends.

Participants will be asked to wear a wrist-worn Axivity AX3 monitor at T1, T3 and T4. Participants will be asked to wear the monitors on a strap on their non-dominant wrist, continuously for seven consecutive days (including when in water and when asleep). Wrist-worn monitors have been validated for use among children and adolescents, in laboratory and free-living environments, and to assess physical activity, sedentary time and postural allocation.[28–30] There is evidence to support the increased acceptability and higher compliance rates of wrist-worn monitors compared with waist-worn monitors.[31–37] To further optimise accelerometer-wear compliance, we have developed a monitor wear and return protocol which is led by researchers (and not teachers) and includes regular reminders and an incentive (eg, GoActive-branded headphones, GoActive-branded pens). We have previously successfully applied this protocol in adolescent cohort studies to obtain high levels of valid accelerometry data (ROOTS: 825/930–89%[7]; SPEEDY-3: 428/480–89%[16]).

Throughout data collection, we will continuously monitor response rates and take appropriate action (eg, requesting teacher involvement) if it drops <70% for the primary outcome. In cases where participants do not return their accelerometer after frequent requests, they may not be issued a monitor at subsequent measurements, but will be allowed to continue their participation in the study and all other (secondary) measures. This is to prevent excessive monitor loss. We deem this appropriate as sample size calculations indicate that we will retain 95% power should retention drop to 55% (80/150 participants predicted to participate in each school based on pilot data).

Once returned, data (continuous waveform data) from the accelerometers will be downloaded. Non-wear time with a minimum duration of 60 min will be removed; the acceleration threshold for identifying non-worn time will be based on visual inspection of the data.[38 39] As we will use a 24-hour protocol, we plan to apply a diurnal adjustment to reduce any bias that may occur if data were not fully representative of a 24-hour period but will also allow full use of the data collected.[40] For any daily analysis, we will set minimum criteria to ensure hours are equally distributed across whole day.[40]

Continuous waveform data will be converted to be comparable to cut-points used previously for ActiGraph accelerometers used to classify time spent sedentary (equivalent to ≤100 ActiGraph cpm) or in light (equivalent to 101–1999 ActiGraph cpm), moderate–vigorous (equivalent to ≥2000 ActiGraph cpm) or appropriate vector magnitude equivalents.[41–43] Monitor output will be reviewed prior to analysis to confirm that these decisions are appropriate for the population and monitor applied. Further, we will consult physical activity measurement experts to ensure we can be aware of relevant new methodology and apply where appropriate. Algorithms to identify sleep time are constantly in development. Given that we are operating a 24-hour wear time protocol, we will use the most up-to-date sleep identification algorithms to remove sleep time when estimating physical activity intensities (particularly sedentary time).

## Anthropometry

Trained staff will measure height, weight and waist circumference following standardised operating procedures (eg, wearing light clothing, removing shoes). Age-specific and sex-specific body fat percentage will be calculated from bio-electrical impedance (collected using Tanita TBF 300 scales), age-specific and sex-specific BMI z-score will be calculated from height and weight. Quality checking of researchers' anthropometry measurements will be conducted prior to baseline measurements and before 10-month follow-up.

## Questionnaires

At each measurement session (ie, T1, T2, T3 and T4), participants will complete a questionnaire concerning secondary outcomes, potential mediators or moderators, and items to monitor any adverse intervention effects. Physical activity type will be assessed using the 30-item Youth Physical Activity Questionnaire, which has previously been validated in 12–17-year-old adolescents.[44] Self-efficacy[45] and social support for physical activity[46] will be assessed using two scales (each with three items). Further items include friendship quality (8-item Cambridge Friendships Questionnaire),[47] well-being (14-item Edinburgh-Warwick Wellbeing Scale),[48] self-esteem (10-item Rosenberg Self Esteem Scale[49] and an adapted social network modelling tool in which participants are provided with a list of tutor group members and asked to select names of their friends),[50] and shyness and sociability (two 5-item measures from EAS temperament scale).[51] Questionnaires will be checked for completion before the end of the measurement sessions, and participants will be asked to complete any missing items. At T1, participants will respond to additional items providing demographic data (ie, age, sex, ethnicity, language spoken at home, parent education and family socioeconomic status). School-level attendance and academic performance (from National Pupil Database) will be collected (publicly available data).

## Process evaluation

Process evaluation will examine the proposed action model for the GoActive intervention (see online supplementary file 1). These process evaluation questions emulate those depicted in Saunders, Evans and Joshi's process-evaluation plan to assess the implementation of a targeted health promotion intervention.[52] We focus on six components: fidelity, dose (delivered and received), reach, recruitment and context.[52–54] See online supplementary file 1 for the applicability and operationalisation of these components.

Intervention process data will include mixed-methods assessment of student, mentor, facilitator, teacher and GoActive staff experiences, and perspectives on intervention delivery, feasibility, acceptance and barriers/facilitators to participation. Reach (eg, the intended amount of students that participate within the intervention) and dose received (eg, the proportion of students who enter points on the GoActive website, how often students download QuickCards and view videos) will be established using the points entries on the study website, download statistics for intervention materials and mentor-reported participation. Process evaluation questionnaires will be administered at T2 and T3 for students (both intervention and control), as well as mentors, facilitators and form teachers in intervention schools. Control participants will be asked to complete process evaluation questionnaires to determine possible contamination. We will include a GoActive logbook for the intervention facilitator and mentors to assess frequency of intervention delivery and any other descriptive notes at T2. Given the flexible, spontaneous and informal nature of the intervention (mentors/leaders attend the same school and can therefore encourage/motivate year 9 students at any time during the week), observation of all intervention delivery is not feasible; but classroom observation will be undertaken to complement other qualitative methods. Existing and emerging school practices which may affect students' physical activity behaviour will be documented and monitored in a structured manner using an adapted school environment questionnaire.[53]

A qualitative researcher will conduct semistructured focus groups, using open-ended questions, after the facilitated intervention phase (T2) with year 9 students in all intervention schools. Approximately 40 students will be selected to participate in the focus groups from all eligible students. Each focus group will comprise approximately four individuals in order to develop themes and generate adequate data. Students will be purposively sampled to ensure a mix of sex, and grouped by level of participation in the GoActive intervention. Subsequent interviews with representatives from all other relevant groups within intervention schools (mentors, teachers and facilitators) will commence in T3. Each focus group (separate for mentors, teachers and facilitators) will comprise 3–8 individuals. An interview guide will be developed and updated as new issues and themes emerge; participants will be encouraged to discuss additional issues. Issues arising will inform the next round of questionnaires and subsequent focus groups, so that additional mechanisms of change can be investigated. In addition to focus groups, individual interviews will be conducted with a purposive sample of inactive and shy Year 9 participants (identified using questionnaire data) at intervention schools to provide a deeper understanding of their intervention experience, and barriers and facilitators to participation (we anticipate these individuals will be more comfortable participating in one-to-one interviews).

At T4, additional semistructured focus groups and interviews with students will explore maintenance of physical activity behaviour change, including who did or did not maintain physical activity behaviour change and why, whether GoActive helped and why or how, and other factors that helped or hindered physical activity maintenance. T2 participants will be reinvited, supplemented by additional students if needed. This gives us a unique opportunity to explore physical activity maintenance across time in the context of a trial and to better understand barriers and facilitators to physical activity maintenance.

## Cost-effectiveness

We will conduct both a within-trial and decision-model-based economic evaluation. The within-trial analysis will be from the cost perspective of the school/local authority. Cost data collected will include intervention-related facilitator time, travel and expenses collected by schools/researchers. Outcomes will comprise change in MVPA and quality-adjusted life-years (QALYs) gained. These will be assessed using the CHU-9D[55] and converted to health state utilities using UK-specific valuations.[56] Change in physical activity observed and costs to schools/local authorities will be input into a previously developed model to predict longer-term costs (to the National Health Service (NHS)) and QALYs hence cost-effectiveness from a public sector perspective (defined as local authority and NHS).

Data collection forms and questionnaires for all measurements are available on request from the corresponding author.

## Data management and monitoring

All data will be collected and managed in line with International Conference on Harmonisation Good Clinical Practice guidelines. Real-time entry and retrospective data validation checks will be conducted. All paper-based questionnaire data will be professionally double data entered and a sample verified for accuracy. Data will be stored securely at the MRC Epidemiology Unit, University of Cambridge, UK. The MRC Epidemiology Unit specialist teams will provide support for training, and quality assessment and control of measurements, and this support will ensure that collection, processing, protection and management of data are timely and of high quality. We will ensure that all provided data are treated as confidential and stored securely. Where this is electronic, data are held on secure computer systems with at minimum password access. All identifiable data will be held on a separate computer system with access limited to appropriate staff by group and password permissions. Personal data will be stored and accessed up to 20 years after study completion.

Due to the low-risk nature of the trial, a formal data monitoring committee has not been appointed. However, the Trial Steering Committee (TSC) will receive regular reports from the investigators and will monitor trial progress and conduct. The TSC will consist of an independent chair, one independent expert, two lay representatives (including a representative from educational sector) and at least two investigators; the committee will be at least 75% independent. The study coordinator and a sponsor representative

will be invited as observers. The TSC will meet approximately once per year, or more frequently if needed. The TSC is responsible for communicating any issues of concern to the sponsor, specifically where the integrity of the study or data or patient safety could be comprised. The study coordinator will also monitor trial conduct and will report independently to the MRC Epidemiology Unit Clinical Research Manager. Potential harms will be monitored by the study team. These will be reviewed by the study coordinator, principal investigator and TSC, and will include reported adverse events (eg, injuries or psychological indicators such as well-being). While we do not expect harm as a result of the GoActive intervention or this trial, it is insured by the University of Cambridge which would provide compensation in case of harm.

The council-funded intervention facilitators will work closely with mentors and research staff to monitor protocol adherence. Poor adherence will be discussed with the research team and TSC, and strategies will be put in place where necessary. No activities are prohibited during the trial as students are expected to do their normal physical activities, including school PE.

Any protocol amendments will be proposed to the TSC and subsequently altered if necessary before submission to funder National Institute of Health Research (NIHR) for approval. Protocol updates will then be uploaded to the NIHR website and trial registry if relevant.

## ANALYSES
### Sample size
We aim to detect a 5 min difference in change in MVPA per day at 10-month follow-up, as observed in the pilot study.[26] A 5 min increase is relevant at population level as it would increase the proportion of adolescents meeting the guidelines of 60 min of MVPA per day from 43% to 50% (based on baseline pilot data), with significant impact on population health.[2] To estimate the required sample size, the following parameters have been used: power=85%, significance level=5%, SD=17.8 (observed in the GoActive pilot),[26] intraclass correlation coefficient=0.034 (observed in SPEEDY-3, n=57 schools),[43] correlation between baseline and follow-up MVPA=0.59 (observed in GoActive pilot, to account for adjustment for baseline MVPA)[26] and average cluster size=100. Based on these parameters, we estimate n=1310 participants will be required for the primary effectiveness analysis. To account for potential school dropout and an estimated loss-to-follow-up of 30%–40%, we aim to recruit 16 schools with 150 participants (total n=2400; average recruitment per school in pilot=154).[26] Should a school have >150 students in year 9, we will include all those who assent to measurement.

### Quantitative analyses
The primary analysis of effectiveness, intermediate and safety outcomes will use an intention-to-treat population, which includes all participants in the group to which they were randomised, regardless of the intervention received. A secondary analysis of efficacy and intermediate outcomes will use a per protocol (PP) population. Inclusion in the PP population will be based on the degree of usage of the intervention website and/or submission of points, and will be defined once clean data are available (but before the start of any trial analyses), when the distributions of degree of website usage can be inspected.

### Outcome analyses
The primary efficacy outcome, MVPA, will be compared between intervention and control groups using analysis of covariance, with adjustment for baseline MVPA; robust SEs will be calculated to allow for the non-independence of individuals within each school. Where baseline values of MVPA are missing, the missing indicator method will be used to enable these participants to be included in the analysis.[57] An estimate of the intervention effect, 95% CI and p value will be calculated. A similar method will be used for the secondary efficacy outcomes. School-level data will also enable analysis of key differences between those participating in the evaluation and the wider school population; for example, patterns of non-response by demographic variables will be explored. Subgroup analyses by prespecified moderators (sex, socioeconomic status, ethnicity, baseline activity level, weight status) will be performed for the primary outcome only. The interaction between randomised group and each moderator will be tested, and if the p value is <0.05, the intervention effect (difference between intervention and control, and 95% CI) will be estimated within each subgroup. The effect on potential mediating variables will initially be assessed as described above. We will subsequently conduct formal mediation analyses using the product of coefficient method[58] to assess the underlying causal pathways of the intervention.

### Qualitative analyses
Focus groups and interviews will be audio recorded, transcribed verbatim and made anonymous. Data will be analysed using thematic analysis following a six-phase model,[59] facilitated by QSR NVivo. Coding will be inductive, incorporating emerging themes as well as topics presented a priori in the interview guide. Initial analyses will inform future data collection and analysis. Interim themes will be discussed by the research team to reach consensus.

### Cost-effectiveness analyses
Cost-effectiveness analyses will follow standardised protocols.[60] The main economic outcome will be the incremental cost-effectiveness ratio, expressed as incremental costs per incremental change in physical activity (MVPA) and per QALY gained (based on CHU-9D) for the trial period (including follow-up). Data collected will include intervention time, travel, expenses, resource use and study-specific costs. In addition, if GoActive increases physical activity, this should reduce adult chronic disease via changes in weight or BMI, and blood glucose. To establish whether GoActive could increase length and/or

quality of life and at what cost, it is not practical to conduct lifetime follow-up, therefore we propose adjusting an existing decision-analytic model to estimate the impact of physical activity on disease risk, quality-adjusted life expectancy (QALY) and cost to the NHS. The modelled analysis will therefore be from a public sector perspective (schools/local authority and NHS).

## Further analyses

Further research questions can be addressed using the cohort data, including (but not limited to) assessment of the predictors of activity maintenance, and the longitudinal association between physical activity/sedentary behaviour and (1) academic performance, (2) shyness and sociability and (3) friendship quality. All proposed analyses will be approved by the project group, and authorship of manuscripts will be informed by recommended guidelines.[61]

## Ethics and dissemination

Ethical approval for the conduct of the study was gained from the Cambridge Psychology Research Ethics Committee, who previously provided ethical approval for the development, feasibility and pilot studies following similar procedures.[22 26]

If successful, it would be appropriate to disseminate this programme to schools and councils across the UK (in addition to peer-reviewed publications). Towards the end of the project, a deliberative dialogue workshop will be held with key stakeholders including students, parents, teachers, school governors and representatives from local/national government. This final workshop will focus on plans for dissemination of results and will include discussion of the process of programme adaptation to a diverse range of secondary schools and further ways of ensuring long-term appeal for adolescents. We anticipate that dissemination could be facilitated through the study website, hosting intervention materials (including videos) and study information.

Given the lack of rigorously evaluated interventions, the results of a CRCT of the effectiveness and cost-effectiveness of GoActive are expected to add substantially to the limited evidence on adolescent physical activity promotion. This study will include an objective, wrist-worn measure of physical activity, aligning with contemporary population surveillance studies[62–64] and ensuring greater protocol compliance for enhanced data retention and quality.[32 34 35] Achieving sustained health behaviour change is an established priority,[10] and so the inclusion of medium-term to long-term follow-up of participants will enable conclusions regarding the trajectories of change (in particular, whether any initial behaviour change is maintained). It will also form one of the largest cohorts in the field of adolescent physical activity promotion, providing many opportunities for secondary data analysis, in addition to testing causal pathways of effect and examining cost-effectiveness. Irrespective of study outcome, the evaluation of the GoActive intervention to increase physical activity in adolescents has the potential for significant academic impact.

**Contributors** The principal investigator KC has overall responsibility for project progress and direction. HB is the day-to-day scientific lead for the project and FW is the operational lead. EvS, PW and AV advise on study procedures and evaluation from their respective disciplines. PW additionally leads the design and evaluation of psychosocial outcomes. CC leads the qualitative and mixed methods research, supported by SJ. SJ leads the process evaluation. EW leads the economic evaluation.

**Funding** This project is funded by the National Institute for Health Research Public Health Research (13/90/18). Intervention delivery costs will be borne by Essex and Cambridgeshire County Councils.

**Disclaimer** Study sponsor and funders have no role in the study design, collection, management, analysis and interpretation of data, writing of the report; and the decision to submit the report for publication.

**Competing interests** None declared.

**Ethics approval** University of Cambridge.

**Provenance and peer review** Not commissioned; externally peer reviewed.

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
