## [Reviewer comments · BMJ Open]

ARTICLE DETAILS

TITLE (PROVISIONAL)	Protocol for a cluster randomised controlled trial to evaluate the effectiveness and cost-effectiveness of the GoActive intervention to increase physical activity among 13-14 year-old adolescents
AUTHORS	Brown, Helen Elizabeth; Whittle, Fiona; Jong, Stephanie; Croxson, Caroline; Sharp, Stephen; Wilkinson, Paul; Wilson, Edward; van Sluijs, Esther; Vignoles, Anna; Corder, Kirsten

VERSION 1 - REVIEW

REVIEWER	Charlotte Edwardson University of Leicester, UK I currently hold a grant that funds a cluster RCT evaluating the effectiveness and cost-effectiveness of a school-based physical activity intervention designed to increase physical activity in 11-14 year olds. This is funded by the same funding stream as the study described in this manuscript i.e., NIHIR PHR
REVIEW RETURNED	14-Oct-2016

GENERAL COMMENTS	This manuscript describes the protocol that will be used in order to test the effectiveness and cost-effectiveness of the GoActive intervention, an intervention designed to increase physical activity in year 9 pupils. This paper is well written and each part of the methodology has been outlined in appropriate detail for other researchers to understand fully what is going to be done and how. The methodology is extremely robust and as such I only have a few minor suggestions and comments. The background section seemed to lack any information on previous school-based interventions that have been conducted. It wasn't clear how this intervention was different to previous school-based interventions. Pg 4, Line 27 – you state 'Further, there is a lack of rigorous evaluation of those existing interventions; for example, in a meta-analysis of 30 studies with objective outcomes, only two of the included studies focused on adolescents over the age of 13 years' The example given here doesn't fit with the start of the sentence i.e., stating that only two studies focused on adolescents over the age of 13 doesn't provide the example of a lack of rigorous evaluation. I would suggest revising to provide a more detailed overview of the lack of RCTs, fully powered studies, using objective measures of PA etc. Performing a process evaluation wasn't mentioned under the objectives.
---

	Its stated that ethical approval will be sought but I assume this has already been granted if baseline data collection is taking place started in September 2016. Pg6, line 12 - You will need 150 participants per school to reach your sample size of 2400 participants from 16 schools. It's likely that schools we will have more than 150 year 9 pupils. Given that with opt-out consent you are likely to have a significant number of pupils eligible to take part how will you select the pupils to be involved in the study if you have more than 150 available? Pg 6, line 21 – how did the parents receive the paper packs – was this through the child? Pg 6, line 35 – update now that ethical approval has been obtained. Pg 7, line 38 – I believe that the Axivity output is similar to the GENEActiv output and therefore you wouldn't get an output of counts per minute – it's likely to be average acceleration. Pg7, line 52 – please state the incentive. Pg8, lines 10-21 – The data processing procedures are appropriate for waking wear ActiGraph data but they don't seem appropriate for use with the Axivity data and for data collected 24 hours/day. Pg8, line 28 – what bioelectrical impedance equipment will you use?
--	---

REVIEWER	Heidi Ruotsalainen Medical Research Center Oulu; University of Oulu, Nursing science and health management, Finland.
REVIEW RETURNED	26-Oct-2016

GENERAL COMMENTS	Thank you for the opportunity to review your manuscript of protocol for a cluster randomized controlled trial to evaluate the effectiveness of the GoActive intervention. Topic is very interesting and important. This protocol is high quality protocol using the SPIRIT statement. All items required, can be found from the manuscript. Authors have done a huge amount of work developing this national wide intervention. Intervention that have been described earlier, is now in evaluation phase. I hope that this CRCT will prove the effectiveness and cost-effectiveness for later implementation phase. I don't have any correction suggestions, except to fig. 1: please add year 2017 to Nov-Jan 2016 Nov 2016-Jan 2017. It would be interesting to read your results from CRCT later. I also appreciate that you'll do a process evaluation, and waiting also to read your results from that. I wish all the best to authors.
---

REVIEWER	Erica Y Lau University of British Columbia, Canada
REVIEW RETURNED	22-Nov-2016

GENERAL COMMENTS	This manuscript described the protocol of a physical activity intervention for adolescents. The intervention is well-designed and the manuscript is well-written. The authors indicate that rigorous evaluation is the strength of this study. I agree that the study has a strong outcome evaluation component, but the process evaluation components are a bit light. As process evaluation information is crucial for researchers to draw valid conclusions on program effectiveness and for future dissemination, I would suggest the authors consider to include a logic model (maybe as supplementary document) to clarify their process evaluation questions and the process indicators: uptake, maintenance and dose need to be better defined.
---

VERSION 1 – AUTHOR RESPONSE

Reviewer 1

This manuscript describes the protocol that will be used in order to test the effectiveness and cost-effectiveness of the GoActive intervention, an intervention designed to increase physical activity in year 9 pupils. This paper is well written and each part of the methodology has been outlined in appropriate detail for other researchers to understand fully what is going to be done and how. The methodology is extremely robust and as such I only have a few minor suggestions and comments.

The background section seemed to lack any information on previous school-based interventions that have been conducted. It wasn't clear how this intervention was different to previous school-based interventions.

This section has been adjusted to briefly reflect the suggested improvements to the GoActive intervention, in comparison with pre-existing programmes (page 3):

“Reviews highlight the limited efficacy of existing adolescent physical activity promotion interventions.^{12,13,14,15} We have previously identified several possible reasons for this lack of effectiveness^[5]; for example, many interventions only target subgroups (such as girls^[6] or low socio-economic groups^[7]) despite activity declining among all groups.^[5] We aim to recruit the whole school year group for evaluation, and to target all groups in the GoActive intervention, which to our knowledge has rarely been done in physical activity promotion interventions. In addition, the decline in activity mainly occurs out of school;¹⁶ however, many interventions only target specific school-based times; for example, school time^{13,20} or Physical Education lessons.²¹ whereas GoActive encourages participants to do more activity both in and out of school.”

Pg 4, line 27 – you state ‘Further, there is a lack of rigorous evaluation of those existing interventions; for example, in a meta-analysis of 30 studies with objective outcomes, only two of the included studies focused on adolescents over the age of 13 years’ The example given here doesn't fit with the start of the sentence i.e., stating that only two studies focused on adolescents over the age of 13 doesn't provide the example of a lack of rigorous evaluation. I would suggest revising to provide a more detailed overview of the lack of RCTs, fully powered studies, using objective measures of PA etc.

We agree that this sentence is unclear, and as such, have adjusted it as below. We have also added further information on gaps in the extant literature (page 3): “Reviews highlight the limited efficacy of

existing adolescent physical activity promotion interventions.^{12,13,14,15} We have previously identified several possible reasons for this lack of effectiveness^[5]; for example, many interventions only target subgroups (such as girls^[6] or low socio-economic groups^[7]) despite activity declining among all groups.^[5] We aim to recruit the whole school year group for evaluation, and to target all groups in the GoActive intervention, which to our knowledge has rarely been done in physical activity promotion interventions. In addition, the decline in activity mainly occurs out of school;¹⁶ however, many interventions only target specific school-based times; for example, school time^{13,20} or Physical Education lessons.²¹ whereas GoActive encourages participants to do more activity both in and out of school. Further, very few adolescent physical activity interventions, especially among older adolescents, have been evaluated using objective measurement of physical activity,¹⁴ and including long-term follow-up, process evaluation, or an assessment of cost-effectiveness.²² This therefore highlights an urgent need for more rigorous evaluation of potentially effective strategies to increase physical activity in adolescents.”

Performing a process evaluation wasn't mentioned under the objectives.

We have added our intention to conduct a process evaluation to the Objectives section (page 3):

Further, we will assess short term (within-trial) and potential long term cost-effectiveness of the GoActive intervention, and will conduct a comprehensive process evaluation including questionnaires, focus groups (with participants, mentors, and teachers), individual interviews, data from intervention logs, and website analytics.

It is stated that ethical approval will be sought but I assume this has already been granted if baseline data collection is taking place started in September 2016.

As previously, this has been updated to reflect that ethical approval has now been granted (page 4):

“Ethical approval for the conduct of the study was granted by the Cambridge Psychology Research Ethics Committee, who previously provided ethical approval for the development, feasibility and pilot studies following similar procedures.”

Pg 6, line 12 - You will need 150 participants per school to reach your sample size of 2400 participants from 16 schools. It's likely that schools we will have more than 150 year 9 pupils. Given that with opt-out consent you are likely to have a significant number of pupils eligible to take part how will you select the pupils to be involved in the study if you have more than 150 available?

We will include all students who assent to be involved in measurement (regardless of school size), and as such, will have no selection process. This is stated in the 'Participants' section (page 5):

“All Year 9 students (13-14 year-olds) in participating schools will be eligible to participate in study measurements.”

We have also added the following sentence to the analysis section of the manuscript to ensure clarity (page 10):

“Should a school have more than 150 students in Year 9, we will include all those who assent to measurement.”

Pg 6, line 21 – how did the parents receive the paper packs – was this through the child?

Students were provided with a paper version of the parent information pack during an introductory assembly conducted by a member of the GoActive team. We have added this information to the manuscript (page 5). This was also paralleled by an electronic copy of all study documents, sent to parents using the school email system:

“These information packs will be distributed to students during an introductory assembly conducted by a member of the GoActive team; students will be asked to take the packs home to their parents. Parents will also be sent duplicate information via email (‘ParentMail’ or the appropriate equivalent system as agreed by the school).”

Pg 6, line 35 – update now that ethical approval has been obtained.

As above, we have updated the manuscript to reflect that ethical approval has now been granted (page 4):

“Ethical approval for the conduct of the study was granted by the Cambridge Psychology Research Ethics Committee, who previously provided ethical approval for the development, feasibility and pilot studies following similar procedures.”

Pg 7, line 38 – I believe that the Axivity output is similar to the GENEActiv output and therefore you wouldn’t get an output of counts per minute – it’s likely to be average acceleration.

We are working closely with the development team at Axivity, in addition to core members of the UK Biobank Physical Activity Validation Study team (housed at the MRC Epidemiology Unit), to ensure that we use the most appropriate output data from the Axivity monitor. In the manuscript, we describe our intention to apply cut-points comparable to those used for ActiGraph accelerometers to the continuous waveform data downloaded to provide meaningful estimates of time spent in different activity intensities. As work to convert Axivity data into a moderate-to-vigorous activity value that is comparable with that derived previously from ActiGraph accelerometers is currently under development; we will use the most up to date algorithms to process this data to enable the Axivity output to be representative of time spent in MVPA. We have edited the manuscript to refer to more appropriate methodology for deriving non-worn time and for defining a valid day from continuous waveform data (page 7):

“Once returned, data (continuous waveform data) from the accelerometers will be downloaded. Non-wear time with a minimum duration of 60 minutes will be removed; the acceleration threshold for identifying non-worn time will be based on visual inspection of the data.[12,13] As we will use a 24-hour protocol, we plan to apply a diurnal adjustment to reduce any bias that may occur if data was not fully representative of a 24 hour period but will also allow full use of the data collected.[40] For any daily analysis, we will set minimum criteria to ensure hours are equally distributed across whole day.[40]

Continuous waveform data will be converted to be comparable to cut-points used previously for ActiGraph accelerometers used to classify time spent sedentary (equivalent to ≤ 100 ActiGraph cpm), or in light (equivalent to 101 - 1999 ActiGraph cpm), moderate-vigorous (equivalent to ≥ 2000 ActiGraph cpm) or appropriate vector magnitude equivalents.[14–16] Monitor output will be reviewed prior to analysis to confirm that these decisions are appropriate for the population and monitor applied. Further, we will consult physical activity measurement experts to ensure we can be aware of relevant new methodology and apply where appropriate.”

Pg 7, line 52 – please state the incentive.

The incentive that is used at each data collection point differs (to ensure students remain interested throughout the measurement programme). To indicate the type of incentives that will be offered, we have added an example to the manuscript (page 6):

“To further optimise accelerometer-wear compliance, we have developed a monitor wear and return protocol which is led by researchers (in collaboration with teachers), and includes regular reminders and a different small incentive after each measurement session (e.g. GoActive-branded headphones, GoActive branded pens).”

Pg 8, lines 10-21 – The data processing procedures are appropriate for waking wear ActiGraph data but they don't seem appropriate for use with the Axivity data and for data collected 24 hours/day.

We will operate a 24-hour wear protocol for the Axivity monitor (see comment above). Algorithms to identify sleep time are constantly in development; we will use the most up to date sleep identification algorithms to remove sleep time when estimating physical activity intensities, especially sedentary time. This issue should not impact on the derivation of our main outcome, MVPA, as any movement occurring during sleep should be at the lower end of the intensity spectrum. It is possible that the Axivity monitor on the wrist may move when sleeping and therefore sleep would become classified as sedentary time (depending on how still the participant is) rather than non-wear which would inflate our estimates of sedentary time (a secondary outcome) if not removed. To ensure this is clear to readers of this manuscript, we have added the following to the relevant section:

“Algorithms to identify sleep time are constantly in development. Given that we are operating a 24-hour wear time protocol, we will use the most up to date sleep identification algorithms to remove sleep time when estimating physical activity intensities (particularly sedentary time).”

Pg 8, line 28 – what bioelectrical impedance equipment will you use?

We have added full details of the bioelectrical impedance scales used to measure body composition to the manuscript (page 7):

“Age- and sex-specific body fat percentage will be calculated from bio-electrical impedance (collected using Tanita TBF 300 scales)...”

Reviewer 2

Thank you for the opportunity to review your manuscript of protocol for a cluster randomized controlled trial to evaluate the effectiveness of the GoActive intervention. Topic is very interesting and important. This protocol is high quality protocol using the SPIRIT statement. All items required can be found from the manuscript. Authors have done a huge amount of work developing this national wide intervention. Intervention that have been described earlier, is now in evaluation phase. I hope that this CRCT will prove the effectiveness and cost-effectiveness for later implementation phase. It would be interesting to read your results from CRCT later. I also appreciate that you'll do a process evaluation, and waiting also to read your results from that. I wish all the best to authors.

I don't have any correction suggestions, except to fig. 1: please add year 2017 to Nov-Jan 2016  Nov 2016-Jan 2017.

Thank you for your attention to detail (and your positive response to the manuscript!); Figure 1 has been adjusted as suggested.

Reviewer 3

This manuscript described the protocol of a physical activity intervention for adolescents. The intervention is well-designed and the manuscript is well-written. The authors indicate that rigorous evaluation is the strength of this study. I agree that the study has a strong outcome evaluation component, but the process evaluation components are a bit light.

As process evaluation information is crucial for researchers to draw valid conclusions on program effectiveness and for future dissemination, I would suggest the authors consider to include a logic model (maybe as supplementary document) to clarify their process evaluation questions and the process indicators: uptake, maintenance and dose need to be better defined.

We agree that including the logic model for the GoActive intervention would be useful. This has been published previously, and is referenced in the Process Evaluation section of the manuscript (page 7). We have also clarified the terms ‘uptake’, ‘maintenance’, and ‘dose’ (page 8):

“Uptake (e.g. how many students participate in GoActive activities), dose (e.g. how often students download QuickCards), and maintenance (e.g. whether students continue to upload points to the website throughout the intervention) will be established using the points entries on the study website, download statistics for intervention materials and mentor-reported participation.”

VERSION 2 – REVIEW

REVIEWER	Erica Lau University of British Columbia, Canada
REVIEW RETURNED	25-Dec-2016

GENERAL COMMENTS	Comments to the authors: The referenced “logic model” (ref #22) is a Change Model but not an Action Model. A change model explains why does the intervention affect the outcomes and this model often includes the intervention component, determinant (mediators) and the outcomes. However, the focus of a process evaluation should be on the Action Model. An Action Model describes how are the contextual factors and program activities are organized for implementing the intervention and supporting the change process. This model usually would describe the intervention activities and their respective process indicators (e.g. dose, responsiveness, reach, quality etc.). Currently, the authors referenced a Change model in the Process Evaluation section, but all the descriptions in this section are related measures for indicators in an Action Model. Here are some references for developing a logic model for process evaluation: Saunders RP, Evans AE, Kenison K, Workman L, Dowda M, Chu YH. Conceptualizing, implementing and monitoring a structural health promotion intervention in an organizational setting. Health Promot Pract 2013;14(3):343-353. Chen H-T. Practical program evaluation: Assessing and improving planning, implementation, and effectiveness. USA: Sage Publications, Inc; 2005. The inconsistency of terminologies is a major challenge in implementation science. It would be useful if the authors could provide a citation for the definitions of uptake, dose, and maintenance. Currently, the process evaluation indicators (i.e., uptake, maintenance and dose) used in this manuscript are not consistent with the literature. The authors’ definitions for uptake (how many students participated), dose (how often students download QuickCards) and maintenance (whether students continues to upload points to the website throughout the intervention) appear to fall under the category of participants’ responsiveness/dose received accordingly to commonly cited literature. Here are some references for the authors to consider when defining the process evaluation indicators: Dane AV, Schneider BH. Program integrity in primary and early secondary prevention: are implementation effects out of control? Clin Psychol Rev 1998;18(1):23-45. Saunders RP, Evans AE, Kenison K, Workman L, Dowda M, Chu YH. Conceptualizing, implementing and monitoring a structural health promotion intervention in an organizational setting. Health Promot Pract 2013;14(3):343-353.
--

VERSION 2 – AUTHOR RESPONSE

Reviewer 3

The referenced “logic model” (ref #22) is a Change Model but not an Action Model. A change model explains why does the intervention affect the outcomes and this model often includes the intervention component, determinant (mediators) and the outcomes. However, the focus of a process evaluation should be on the Action Model. An Action Model describes how are the contextual factors and program activities are organized for implementing the intervention and supporting the change process. This model usually would describe the intervention activities and their respective process indicators (e.g. dose, responsiveness, reach, quality etc.). Currently, the authors referenced a Change model in the Process Evaluation section, but all the descriptions in this section are related measures for indicators in an Action Model. Here are some references for developing a logic model for process evaluation:

Saunders RP, Evans AE, Kenison K, Workman L, Dowda M, Chu YH. Conceptualizing, implementing and monitoring a structural health promotion intervention in an organizational setting. *Health Promot Pract* 2013;14(3):343-353.

Chen H-T. *Practical program evaluation: Assessing and improving planning, implementation, and effectiveness*. USA: Sage Publications, Inc; 2005.

After appraisal of the literature suggested, we have added a supplementary file detailing the process evaluation, including fidelity, dose delivered, dose received, reach, recruitment, and context of the GoActive intervention.

The inconsistency of terminologies is a major challenge in implementation science. It would be useful if the authors could provide a citation for the definitions of uptake, dose, and maintenance. Currently, the process evaluation indicators (i.e., uptake, maintenance and dose) used in this manuscript are not consistent with the literature. The authors' definitions for uptake (how many students participated), dose (how often students download QuickCards) and maintenance (whether students continues to upload points to the website throughout the intervention) appear to fall under the category of participants' responsiveness/dose received accordingly to commonly cited literature. Here are some references for the authors to consider when defining the process evaluation indicators:

Dane AV, Schneider BH. Program integrity in primary and early secondary prevention: are implementation effects out of control? *Clin Psychol Rev* 1998;18(1):23-45.

Saunders RP, Evans AE, Kenison K, Workman L, Dowda M, Chu YH. Conceptualizing, implementing and monitoring a structural health promotion intervention in an organizational setting. *Health Promot Pract* 2013;14(3):343-353.

We have adjusted the identified process evaluation terminology as advised. Please see Process Evaluation section, and the additional Supplementary File.

Additional adjustments

After discussion with our Trial Steering Committee, we have removed the anthropometric measurement from T3 (the post-intervention follow-up). This is to reduce the measurement burden on participating students and ease the organisation of sessions for contact teachers in our recruited schools. We do not expect immediate changes in weight status, waist circumference, or body fat percentage post-intervention. Figure 1 has been adjusted to reflect this change, and added the following text to the manuscript to clarify:

All primary and secondary outcomes will be assessed at T1 and T4. Anthropometric measures will be removed from T3 (which will include all other outcomes, i.e. accelerometry and questionnaire-based measures), and T2 will focus on assessing the questionnaire-based measures only (including mediators of change).

VERSION 3 – REVIEW

REVIEWER	Erica Lau University of British Columbia, Canada
REVIEW RETURNED	07-Mar-2017

GENERAL COMMENTS	The authors have appropriately addressed the previous comments by clarifying the definitions of the process evaluation indicators and the links between the indicators and their measures. My only suggestion for this revision is to revise the measures for Fidelity. Currently, the authors included student website usage (“google analytics on frequency and duration of website use, resources download and points upload statistics) as one of the measures for fidelity. However, this is a measure for dose received. Fidelity should reflect the quality of program implementation, which usually determined based on how well the essential elements were demonstrated. According to the GoActive theoretical model (ref #22), the essential elements of this intervention could be defined as encouragement and modeling from mentors and leaders, and incentivization. I would suggest the authors include a few items to measure these aspects.
---

VERSION 3 – AUTHOR RESPONSE

Reviewer: 3

Reviewer Name: Erica Lau

Institution and Country: University of British Columbia, Canada

Please state any competing interests: None declared

The authors have appropriately addressed the previous comments by clarifying the definitions of the process evaluation indicators and the links between the indicators and their measures. My only suggestion for this revision is to revise the measures for Fidelity. Currently, the authors included student website usage (“google analytics on frequency and duration of website use, resources download and points upload statistics) as one of the measures for fidelity. However, this is a measure for dose received. Fidelity should reflect the quality of program implementation, which usually determined based on how well the essential elements were demonstrated. According to the GoActive theoretical model (ref #22), the essential elements of this intervention could be defined as encouragement and modeling from mentors and leaders, and incentivization. I would suggest the authors include a few items to measure these aspects.

Response

Amendments have been made to the discussion of fidelity in Supplementary File 1. Student website usage will be indicative of both dose received and fidelity of the intervention. Student website usage will provide the authors an understanding of whether the intervention was implemented as per protocol (e.g. whether the students logged points as per protocol, and hence, will also provide a measure as to the success of the incentivization). However, in response to the reviewer’s comment, we have removed Google analytics on frequency and duration of website use, and resources download has been removed as we agree they do not specifically measure fidelity. Fidelity of

intervention implementation will be assessed utilizing an observation procedure. This will include observing the encouragement and modelling of activities from mentors and leaders to Year 9 students.

VERSION 4 – REVIEW

REVIEWER	Erica Lau University of British Columbia, Canada
REVIEW RETURNED	30-Mar-2017
GENERAL COMMENTS	The authors have addressed all my comments.